# What Does Provide Better Effects on Balance, Strength, and Lower Extremity Muscle Function in Professional Male Soccer Players with Chronic Ankle Instability? Hopping or a Balance Plus Strength Intervention? A Randomized Control Study

**DOI:** 10.3390/healthcare10101822

**Published:** 2022-09-21

**Authors:** Hadi Mohammadi Nia Samakosh, João Paulo Brito, Seyed Sadredin Shojaedin, Malihe Hadadnezhad, Rafael Oliveira

**Affiliations:** 1Department of Biomechanics and Corrective Exercises and Sports Injuries, University of Kharazmi, Tehran 15719-14911, Iran; 2Sports Science School of Rio Maior–Polytechnic Institute of Santarém, 2040-413 Rio Maior, Portugal; 3Life Quality Research Centre, 2040-413 Rio Maior, Portugal; 4Research Center in Sport Sciences, Health Sciences and Human Development, 5001-801 Vila Real, Portugal

**Keywords:** dynamic balance, isometric, jump, stability, static balance

## Abstract

Chronic ankle instability (CAI) has a higher frequency in soccer due to the rapid changes in body movement. Thus, this study compared the effects of eight weeks of a hopping protocol and a combined protocol of balance plus strength in a within-between group analysis. Thirty-six male professional soccer players participated in this study and were randomly allocated in three groups: control group (CG, *n* = 12), hopping group (HG, *n* = 12), and balance plus strength group (BSG, *n* = 12). Strength, static and dynamic balance, and function were assessed at baseline and eight weeks post intervention. First, Foot and Ankle Ability Measure (FAAM) and FAAM sport scales were applied. Then, a dynamometer was used to measure strength of the muscles around the hip, knee, and ankle joints. The Bass stick measured static balance and the Y balance test measured dynamic balance. Additionally, functional tests were carried out by Triple Hop, the Figure 8 hop, and vertical jump. A repeated measures ANOVA [(3 groups) × 2 moments] was used to compare the within and between group differences. In general, all tests improved after eight weeks of training with both protocols. Specifically, the BSG improved with large ES for all tests, while the HG improved all test with small to large effect sizes (ES). Furthermore, HG showed higher values for vertical jump (*p* < 0.01, ES = 1.88) and FAAMSPORT (*p* < 0.05, ES = 0.15) than BSG. BSG showed higher values for hip abduction (*p* < 0.05, ES = 2.77), hip adduction (*p* < 0.05, ES = 0.87), and ankle inversion (*p* < 0.001, ES = 1.50) strength tests, while HG showed higher values for knee flexion [ES = 0.86, (0.02, 1.69)] and ankle plantarflexion [ES = 0.52, (−0.29, 1.33)]. Balance plus strength protocol showed more positive effects than the hopping protocol alone for soccer players with CAI.

## 1. Introduction

Chronic ankle instability (CAI) is characterized by a wide spectrum of long-term sensorimotor and mechanical deficits that cause recurrent damage to the lateral ankle-ligament complex, and consequently, a subjective feeling of ankle-joint instability during functional or dynamic movements [1]. CAI can be caused by lateral ankle sprain, which is one of the most common injuries during exercise and daily living activities [2,3,4]. This type of injury accounts for 15 to 30 percent of sports injuries and has a return rate to training of ~75 percent [3]. Generally, injuries such CAI occur more frequently in soccer, futsal, basketball, volleyball, and sports that require a rapid change in direction [5]. Consequently, it is known that one of the most important lower extremity joints that helps maintain one’s center of mass and body control stability is the ankle [6]. Postural control has both static and dynamic dimensions. The effort to maintain a high level of support with the least amount of movement is called static posture control, and the attempt to maintain a high level of support while performing several movements is called dynamic posture control [7,8]. As one of the controversial concepts of the sensory-motor system, the topic of postural control explores the complex and Interrelated relationship between sensory inputs and the motor responses required to maintain or modify the posture [5].

An ankle injury could be related to various mechanical receptors, such as the articular capsule, ligaments, and tendons of the ankle joint [2]. When afferent inputs change after injury, muscle contractions change, and therefore, mechanical receptor injuries can lead to functional and balance issues, deficits in strength, and chronic instability [2,5]. However, several exercise training programs, such as balance, Pilates, core stability, aquatic therapy, and proprioception exercise, have been shown to improve static and dynamic balance of people with CAI [2,5]. Despite evidence on improving muscle strength, there has been conflicting research concerning effects on proprioception of balance [9]. Another training approach that recently showed improvements in athletes with CAI is the use of hopping protocols [10,11]. Ardakani et al. [11] showed that a six-week hopping protocol improved Foot and Ankle Ability Measures (FAAM), Cumberland Ankle Instability Tool (CAIT), Foot and Ankle Outcome Score, Peak Ground Reaction Forces, Time to Peak Ground Reaction, and several movements of the hip, knee, and ankle in male university basketball players with CAI [11]. Minoonejad et al. [10] also showed improvements in FAAM, CAIT, and Foot and Ankle Outcome Scores, as well as preparatory muscle activation, reactive muscle activation, and muscle onset time across the lower extremity with six weeks hopping protocol. However, none of the previous studies analyzed static and dynamic balance [10,12]. The literature is scarce on the effects of isolated plyometric training and those that combine balance plus strength training on neuromuscular adaptation on chronic ankle instability in soccer players. On the one hand, ankle sprain is common among soccer players; on the other hand, dynamic and static balance are essential to the soccer game. Therefore, the main aim of the present study was to compare the effects of two exercise training programs (a hopping protocol and a combined protocol of balance plus strength) in professional soccer players. The primary hypothesis was that both training protocols can improve function, balance, and strength of lower extremity muscles in soccer players with CAI. The secondary hypothesis was that balance plus strength would provide better effects than a hopping protocol.

## 2. Materials and Methods

### 2.1. Design

The present study is a randomized clinical trial with an IRCT code of IRCT20190830044643N1, with control group, parallel groups, and two blinding testers (certified strength and conditioning professionals).

### 2.2. Participants

This study was conducted with men professional soccer players suffering from CAI in Iran. Thirty-nine players were selected from two first-division professional clubs (Azadegan league) across the country.

A questionnaire was used to collect information about exercise history, and CAI was diagnosed by the team physician, who has a sports medicine degree, following the International Ankle Consortium, which means that for an injury to be considered as CAI, the following inclusion criteria were adopted [13]: at least one severe ankle sprain in the past; the first sprain must have happened at least 12 months before enrolling in the trial; the initial sprain has to be accompanied by inflammatory signs (i.e., pain, edema, and so on); at least one day of targeted physical activity must have been disrupted by the first injury; the most recent injury must have occurred at least three months before enrolling in the study; previous ankle joint injury that gives way, recurrent sprain, or feelings of instability; in the six months leading up to study enrolment, participants must report at least two occurrences of their ankle giving way; experiencing a recurring sprain, which is defined as two or more sprains on the same ankle; instability in the ankle joint; Cumberland Ankle Instability Tool < 27, which confirmed self-reported ankle instability [14]. Additionally, the following exclusion criteria were applied: history of past musculoskeletal surgery in either lower extremity limb; history of a fracture in one of the lower extremity limbs that necessitated a realignment; acute damage to the musculoskeletal structures of other joints in the lower extremity in the past three months that resulted in at least one day of missed physical activity; ankle anterior drawer test was positive; only one training session was missed.

The selection of subjects with CAI was conducted through a clinical anterior drawer test of ankle, and scores were obtained using the CAI questionnaire (i.e., Cumberland Ankle Instability Tool) [13,14]. The questionnaire, which consists of nine multiple-choice questions, provides information on the extent and onset of pain and symptoms associated with functional ankle instability. A lower score indicates greater severity of the injury as experienced by the person answering. The validity and reliability of this questionnaire in assessing the severity of ankle sprain injury have been confirmed by previous studies [15]. The functional stability score of the ankle is between 0 and 30; a score range of 27 to 30 represents good health of the ankle, while a score between 0 and 27 indicates more severe ankle instability [16]. The following eligibility criteria to participate in the training protocols were applied. Inclusion criteria: (1) players had unilateral ankle instability (dominant leg); (2) no musculoskeletal abnormalities of the lower extremity and no lower back pain; (3) players had to complete a minimum of 85% of training sessions; (4) no injuries in the last three months. Exclusion criteria: (1) absence in three training sessions and/or tests; (2) stop following the intended exercise protocol.

Before staring the study, players and their parents signed the informed consent form to be examined according to the ethical principles contained in the Declaration of Helsinki (2013), and they declared their voluntary participation. Ethical permission was given by the Human Ethics Committee at the University of Tarbiat Modares Iran code of IR.MODARES.REC.1397.043.

### 2.3. Sample Size

In order to achieve sample power, a priori F-test family, with the statistical test of ANOVA: repeated measures, within-between interaction (α = 0.05, 1 − β = 0.95 and d: 0.5), through G*Power software (RRID:SCR_013726) was used [17]. A minimum of 21 subjects should be included in the analysis to achieve 97% of actual power. Thus, a total of 39 players were selected to this study. Three players did not complete the assessments. All players were committed to finish the training session unless they were injured. In the present study, three players withdrew from the intervention due to injuries, such as an injury in a within-team match in the second week (two players in control group). In addition, one player from BSG did not participate in the post test. Therefore, 36 players were considered for further analysis into the three groups: hopping group (HG), BSG, and control group (CG).

### 2.4. Randomization

A blinded person generated the allocation sequence, and two other researchers as trainers enrolled players and assigned them to interventions. The allocation sequence was random. Additionally, the allocation sequence was concealed until players were enrolled and assigned to interventions.

Players were assigned to one of three groups with an equal randomization (ratio of 1:1:1) (Figure 1). Randomization was performed by a blinded person who did not know the aims or design of the study. Randomization was performed with each person-specific naming code (up to 39) previously sealed: the blind person was asked to place 13 cards in three balls [18].

### 2.5. Procedures

All players performed a routine of exercises planned by the coaches and certified strength and conditioning professionals. Prior to the tests, the participants executed a standardized 5-min warm-up protocol consisting of a series of double leg squats (2 × 8 repetitions) and double leg maximum jumps (2 × 5 repetitions), followed by dynamic calf-stretching with a straight and bent knee. The intervention groups performed three additional training sessions 1–2 h before the start of their routine training per week (45–60 min, each session) accordingly to the group attributed, while the CG only performed the regular soccer training sessions. The players who were in intervention groups rested between 30–60 min until starting the regular soccer sessions. These sessions were held on Mondays, Wednesdays, and Fridays and were composed of speed endurance sessions (e.g., long sprints, repeated sprints), aerobic high-intensity sessions (e.g., interval training, medium-to-large sized games), and ball-possession games or team/opponent tactics sessions, respectively. On Tuesdays, Thursdays, and Saturdays, all players rested. On Sundays, they played one within-team match per week (intra-team friendly match).

The intervention occurred in the pre-season and lasted eight weeks. The intervention groups were assessed one week before the start of training and one week after, while the CG was assessed one week before and two weeks after the training programs finished.

Before intervention protocols, demographic data and exercise history were collected. Static and dynamic balance were assessed using Bass-Stick and Y-balance tests (Move 2 Perform, Evansville, IL, USA), respectively [19,20]. Muscle strength was assessed by a handheld manual dynamometer (MMT) (North Coast Made in USA) [21]. Lower extremity function was assessed through the Functional Indices of Triple Hop test, Figure 8 Hop and vertical jump tests. Additionally, FAAM and FAAMSPORT disability questionnaires were used [22]; it should be noted that all evaluations were performed with the dominant leg, which was specified using the ball kicking test [23]. The test-retest reliability was clinically acceptable for FAAM (intraclass correlation, ICC = 0.87) and FAAMSPORT (ICC = 0.87) [22]. Both questionnaires included several questions regarding ankle pain and function, including the FAAM Activities of Daily Living (ADL) and Sport scales. The FAAM comprises two scales: the ADL subscale and the Sport subscale. The ADL subscale consists of 21 questions that pertain to various functional activities one would encounter in normal daily activity. The Sport subscale contains eight questions pertaining to different activities that are related to sport participation. Participants were asked to rate the activity as no difficulty at all (4 points), slight difficulty (3 points), moderate difficulty (2 points), extreme difficulty (1 point), unable to do (0 points), or N/A (not applicable). The FAAM scores were recorded as a percentage of 84 points [22].

#### 2.5.1. Assessment of Dynamic Balance

To begin the Y-balance test, the actual lower limb length from the anterior superior iliac spine to the medial was measured to normalize the data and compare subjects. To measure leg length, the subject was first asked to lie on the back of the table, then the distance between the anterior superior iliac spine to the distal portion of the medial ankle malleolus was repeated twice for each subject and each leg. The mean is then calculated as the leg length. The subject stood in the center of the room, then rested on one leg and then on the other leg, returned to normal position on two legs, and remained in this position for 10 to 15 s before the next attempt [20]. Three efforts in one direction had to be completed before moving in the other direction and had to be performed in a sequential clockwise or counterclockwise manner. The subject’s toe touched the highest possible point in the specified directions, while the distance between the contact point center and touched point was measured in centimeters. The soccer player cannot touch their foot down on the floor before returning to the starting position. Any loss of balance will result in a failed attempt. However, once they have returned to the starting position, they are permitted to place their foot down behind the stance foot. The soccer player cannot place their foot on top of the reach indicator to gain support during the reach—they must push the reach indicator using the red target area. The soccer player must keep their foot in contact with the target indicator until the reach is finished. They cannot flick or kick the reach indicator to achieve a better performance [20]. Each participant completed 2–3 trials to get familiarized. The rest time was 30 s between three efforts, and the rest time was 60 s when changing to the other direction.

The following formula is used to determine the mean balance scores (Y-balance test) in each direction: total score = Y balance–Anterior (cm) + Y balance–Posteromedial (cm) + Y balance–Posterolateral (cm)/3. The score was expressed in centimeters [20].

#### 2.5.2. Isometric Tests

##### Assessment of Isometric Hip Abductor Strength

Hip abductors were measured in a position at the examination table, with a strap at the desired angle (the hip abducted 10 degrees to maintain normal position). The subject’s trunk was fixed using a strap attached to the iliac crest superior and around the table. In the tested leg, the center of pressure of the dynamometer was placed at a point 5 cm proximal to the lateral line of the knee joint. After zeroing the isometric dynamometer (North Coast Medical Inc, CA, USA) [21], the subject was requested to perform a maximal isometric contracture of the hip and to hold it for 5 s three times with an interval of 15 s [24]. The rest time was 60 s between three trials, and the rest time was 120 s between each of the isometric tests.

##### Assessment of Isometric Hip Adductor Strength

The hip adductor strength was measured when the subject was lying sideways on the examination table, while the hip joint was tested in extension and while it was flexed to 90°. The subject with the upper hand grasped the edge of the table, and the lower side was in a comfortable position under the subject. When the dynamometer was positioned 5 cm above the femur’s medial condyle, the subject was required to perform a maximal contracture and to hold it for 5 s, three times, with an interval of 15 s between each test [24].

##### Assessment of Isometric Knee Flexor Strength

Knee flexor strength was measured in a sitting position on the table with the knee positioned at 90° relative to the femur, while the femur was fixed with two bands. The position of the head dynamometer is considered to measure the flexor force on the knee, the surface of the distal of the posterior leg. After zeroing the dynamometer, the subject was requested to perform a maximal isometric contracture of the knee flexion and to hold it for 5 s. This test was performed three times with an interval of 15 s between each test [24].

##### Assessment of Isometric Knee Extensor Strength

Knee extensor strength measured in a sitting position on a table with the knee positioned at 90° relative to the hip while the femur was fixed with two bands. The position of the head dynamometer is considered to measure the extensor force on the knee and the surface of the distal of the anterior leg. After zeroing the dynamometer, the subject was requested to perform a maximal isometric contracture of the knee extension and to hold it for 5 s. For this test, three trials with an interval of 15 s between each test were performed [24].

##### Assessment of Isometric Ankle Plantarflexion and Dorsiflexion Strength

The subject was in a sitting position with the knee joint extending to the ankle at 0°. To measure plantar strength, a dynamometer was positioned on the proximal metatarsophalangeal surface on the palmar surface while lowering the ankle, and to measure dorsiflexor strength. After zeroing the dynamometer, the subject was requested to perform a maximal isometric contraction of the ankle plantarflexion and dorsiflexion and to hold it for 5 s. For this test, three trials with an interval of 15 s between each test were performed [24].

##### Assessment of Isometric Ankle Inversion and Eversion Strength

Inversion and eversion ankle strength was measured in a lying-back position with the legs extended outward and forward. After zeroing the dynamometer, the subject was requested to perform a maximal isometric contraction of the ankle inversion and eversion and to hold it for 5 s. For this test, three trials with an interval of 15 s between each test were performed [24].

For all isometric tests, the average of every three isometric contractions performed was recorded in kg.

#### 2.5.3. Assessment of Triple Hop

This test required a narrow measuring tape that was 6 m long, securely positioned on the ground (Figure 2). The test involved performing three consecutive hops by traveling the maximum distance possible and landing on the same foot in each hop, and ultimately maintaining the landing mode for at least 3 s. Hand movements can be used to maintain balance. After performing two or three attempts, the subject does a triple hop two times for the dominant leg, and the total distance traveled is recorded [25]. The rest time was 60 s between three trials, and the rest time was 120 s between each of the hop/jump test.

#### 2.5.4. Assessment of Figure 8 Hop

The Figure 8 Hop test was used to measure power, speed, and balance of the lower extremity, emphasizing one leg control. The path is 5 m long and 1 m wide, with seven obstacles (three obstacles at the top, three obstacles at the bottom, and one obstacle in the center). The subject stood behind the starting line with his dominant leg, while his other leg was slightly bent from the knee and hip joints. The subject was then instructed to hop at maximum speed to travel the specified path twice, and the elapsed time was recorded with a precision of 0.01 s. The subject was requested to hold his hands on the iliac crest during the test to avoid oscillating hand movements. It is worth mentioning that the subject performed this trial with shoes. The subject performed one to three experimental attempts, with a rest interval of 30 s. If the subjects lost balance during the test or something went wrong, the test was considered an error, and the test was repeated [25].

#### 2.5.5. Assessment of Vertical Jump

The vertical jump test was performed so that the subject would first stand on the dominant leg wall (the location of the foot on the ground was predetermined). Then, with a stretched-out hand, the wall was marked to indicate the initial height. Then, the subject jumped to their maximum, and a sign was put on the wall. This jump was performed three times. Then, the distance between the base mark and the final mark in each jump was measured and recorded as the subject’s vertical jump height [26].

### 2.6. Interventions

#### 2.6.1. Balance plus Strength Protocol

The balance plus strength training protocol consisted of balance and strength types of exercises (examples in Figure 3) that were performed over eight weeks (three sessions per week), 45–60 min per session. Additionally, the intensity of exercise increased, with an increasing number of sets and repeated movements, every two weeks. Specifically, the intensity of exercise increased (sets and repetitions) by observing the principle of overload for each person. Rest between sets lasted 30 s and rest between exercises lasted 1 min (Table 1) [7]. Before and during the balance and exercises, the players were instructed to focus their strength to hold proper alignment of their body, to maintain a visual point on the front floor, to ensure the full alignment of the body, to not bend forward from the waist while undertaking a single leg squat and squat, to not drop the pelvis on the “bridge to plank”, and to raise your knees as much as possible in the tuck jump.

#### 2.6.2. Hopping Protocol

The protocol consisted of six types of hopping exercises (examples in Figure 4) performed for eight weeks, three sessions per week, and 30 min per session. The intensity of the exercises was progressively increased by adding exercises and sets. Table 2 present the Hopping training protocol [27]. Rest between sets lasted 30 s and rest between exercises lasted 1 min. During training, soccer players were given feedback on postural control to avoid injury. They were also instructed on the position of the arms (hands-free, hands behind, or hands-on chest), “keep the focus”, and “feel the foot contact with the ground, avoiding a valgus overall lower limb alignment on landing, and reducing ground reaction forces”.

#### 2.6.3. Statistical Analysis

Data were analyzed using Statistical Package IBM SPSS Statistics for Windows, version 22.0 (IBM Corp., Armonk, NY, USA). A Shapiro–Wilk test was used to assess the normal distribution of data. Then, Levene’s test evaluated the equality of variances to verify homogeneity. Data were presented as means and standard deviations (SDs). Repeated measures ANOVA analysis with partial eta squared (ηp^2^) and the Bonferroni adjustment post hoc test were used to compare pre-post interventions and groups [(3 groups) × 2 moments]. The significance level was set at *p* ≤ 0.05. In addition, relative changes (%) and Cohen’s D effect size (ES) with confidence intervals (CI, 95%) were calculated to measure the magnitude effects. Cohen’s D method suggested that d = 0.2 is considered a ‘small’ effect size, d = 0.5 is considered a ‘medium’ effect size, and d = 0.8 is considered a ‘large’ effect size [28].

## 3. Results

Characteristics of the study are reported in Table 3 for each group at baseline. There were no significant differences between the groups, all *p* > 0.05.

The outcomes of the two-way ANOVA and the Bonferroni post hoc test regarding balance and function tests are presented in Table 4. The ANOVA analysis group x time provided the following significant results: static balance Bass-stick test (F_2.33_ = 53.06, ηp^2^ = 0.82, *p* < 0.001), Y balance–Anterior (F_2.33_ = 69.70, ηp^2^ = 0.84, *p* < 0.001), Y-balance–posteromedial (F_2.33_ = 30.39, ηp^2^ = 0.73, *p* < 0.001), posterolateral (F_2.33_ = 6.84, ηp^2^ = 0.38, *p* = 0.005), total scores (F_2.33_ = 60.80, ηp^2^ = 0.84, *p* < 0.001), triple single leg hop test (F_2.33_ = 4.45, ηp^2^ = 0.28, *p* = 0.02), Figure 8 hop (F_2.33_ = 4.07, ηp^2^ = 0.27, *p* = 0.03), vertical jump (F_2.33_ = 71.90, ηp^2^ = 0.86, *p* < 0.001), FAAM (F_2.33_ = 5.32, ηp^2^ = 0.32, *p* = 0.01), and FAAMSPORT (F_2.33_ = 69.87, ηp^2^ = 0.86, *p* < 0.001).

Regarding pre to post comparisons, BSG showed significant improvements in static balance Bass-stick test, Y balance–Anterior, Y-balance–posteromedial, posterolateral, and total scores, triple single leg hop test, Figure 8 hop, vertical jump, FAAM and FAAMSPORT, while HG showed significant results in Y-balance–posteromedial, posterolateral, and total scores, Figure 8 hop, vertical jump, FAAM and FAAMSPORT. CG did not present significant changes.

Regarding comparisons between groups, BSG presented higher values for Figure 8 hop, vertical jump, FAAM and FAAMSPORT than CG. HG presented higher values for triple single leg hop and Figure 8 hop than CG. Finally, when comparing intervention groups, HG showed higher values for vertical jump [ES = 1.88, (0.93, 2.93)] and FAAMSPORT [ES = 0.15, (−0.65, 0.95)] than BSG. No other significant differences were noted.

Figure 5 also highlights some of the main results for Y balance–total, triple single leg hop, vertical jump and FAAMSport.

The outcomes of the two-way ANOVA and the Bonferroni post hoc test regarding strength tests are presented in Table 5. The ANOVA analysis group × time provided the following significant results: hip abduction (F_2.33_ = 26.84, ηp^2^ = 0.71, *p* < 0.001), hip adduction (F_2.33_ = 16.15, ηp^2^ = 0.59, *p* < 0.001), knee flexion (F_2.33_ = 25.56, ηp^2^ = 0.70, *p* < 0.001), knee extension (F_2.33_ = 14.67, ηp^2^ = 0.57, *p* < 0.001), ankle plantarflexion (F_2.33_ = 9.12, ηp^2^ = 0.45, *p* = 0.001), ankle dorsiflexion (F_2.33_ = 16.91, ηp^2^ = 0.60, *p* < 0.001), ankle eversion (F_2.33_ = 15.29, ηp^2^ = 0.58, *p* < 0.001) and ankle inversion (F_2.33_ = 36.82, ηp^2^ = 0.77, *p* < 0.001).

Regarding pre to post comparisons, BSG showed significant increases in all strength tests, while HG showed significant improvements in hip abduction, hip adduction, knee flexion, knee extension, ankle plantarflexion, ankle dorsiflexion, ankle eversion, and ankle inversion.

Regarding comparisons between groups, BSG presented higher values for hip abduction, hip adduction, knee flexion, knee extension, ankle plantarflexion, ankle dorsiflexion ankle dorsiflexion, and ankle inversion than CG. HG presented higher values for knee extension, knee flexion, ankle plantarflexion, ankle dorsiflexion, and ankle inversion than CG. Finally, when comparing intervention groups, BSG showed higher values for hip abduction [ES = 2.77, (1.67, 3.87)], hip adduction [ES = 0.87, (0.04, 1.71)], and ankle inversion [ES = 1.50, (0.48, 2.52)], while HG showed higher values for knee flexion [ES = 0.86, (0.02, 1.69)] and ankle plantarflexion [ES = 0.52, (−0.29, 1.33)].

## 4. Discussion

In the present study, we analyzed two types of hopping and balance plus strength protocols on balance, strength, and lower extremity strength muscle function in soccer players with CAI. The overall results of the present study showed positive effects on those variables which confirmed our first study hypothesis. Specifically, BSG improved with large ES for all tests, while HG improved all test with small to large ES, which confirmed our second study hypothesis. The results showed higher values for static and dynamic balance, as well as strength in the group that performed BSG protocol, which could be associated when more emphasis is given to the neuromuscular training, which consequently contribute to a greater effectiveness of this type of training protocol. Another justification could be the isolated focus on proprioception and muscle strength components in BSG exercises compared to HG exercises.

Research has shown that muscle weakness and subsequent increased ankle joint loosening and motor-sensory deficits resulting from the sprain are associated with balance deficits and postural control [3]. Moreover, it has been shown that muscle weakness and subsequent increased ankle joint laxity and motor-sensory deficits resulting from the sprain are associated with balance deficits and postural control [3].

In this sense, two training protocols (combined training of hopping and core stability and hopping training alone) provided more desirable outcomes than core stability training [29]. However, the same authors did not find any significant difference between the HG that involved the ankle area and the combined protocol (core stability plus hopping) that included the trunk area in addition to the ankle section. The results of the present study corroborated the previous findings considering the positive results and the differences in the protocols.

A research paper examined the effect of 6 weeks of balance training and combined (balance and plyometric) training using the time to stabilization (TTS) test on the force plate and balance by the one-leg stand test. The results showed that combined training (Balance-Plyometric) more effectively reduced postural oscillations in static and dynamic conditions compared to balance training alone [30].

To justify the greater effectiveness of BSG, it seems that a combination of many types of exercises together improved the balance and strength of soccer players with CAI. The study of Hall et al. [30] examined the effects of two types of BSG on balance, strength, and lower extremity function in people with CAI and showed that both kinds of training improved those variables [30], which is also supported by the results of the present study.

Exercise interventions on hip and lower extremity strength among players should be considered because lack of control in these areas will lead to knee valgus dynamic [31]. In this regard, it is stated that the strength of the hip joint is essential for proper walking mechanics and foot posture when heel contact impact changes postural stability and muscle strength absorption patterns in the hip and ankle after injury, which can be effective in reinjury in the future [32]. As stated by the authors, there are training protocols that improve the strength and endurance of the core muscles that can explain the effect of enhancing balance so that the core muscles create a solid cylinder, and subsequently, produce inertia against body perturbation, providing the body with a movement-stable surface. The abdominal muscles, including the transverse abdominis, rectus abdominis, medial oblique, and lateral oblique, are all integrated to provide spinal stability and, thus, a stronger support surface for lower extremity movement [32].

In some cases, the importance of proper activation and stability of the trunk while maintaining static stance control has also been emphasized [33]. Kibler et al. [34] described that the activation of the central muscular structure of the body, in patterns associated with limb movement, contributes to the development of balance and subsequent function. According to the same authors, the body triggering the core muscles used to yield the necessary rotational torque around it and to produce limb movement [34]. The second part of the BSG, balance and proprioception training, can have a greater impact on training than hopping training. As stated, injury can have a profound effect on proprioception and neuromuscular control. Munn et al. [35], in a review article on individuals suffering CAI, reported that secondary postural defect was due to control, neuromuscular, and proprioception deficits [35]. The outcomes of the present study showed that BSG protocol was significantly more effective in improving Y-balance test in all directions than the results of other studies that examined the effect of balance training. Hall et al. also applied a balance training protocol in the anterior, posteromedial, and posterolateral direction and reported significant improvements [2].

Researchers have claimed that hopping training creates a link between strength and coordination and directly enhances competitive performance [36]. The present study showed positive effects of the hopping protocol on the static and dynamic balance. This is in line with the outcomes reported by Myer et al. [37] through the application of a plyometric training program [37]. Therein, Kramer et al. [38] pointed out the importance of using this kind of training to maintain muscle strength and stated that using plyometric training may prevent the neuromuscular dysfunction of people with low activity and people with impaired posture due to prolonged sitting. In addition, balance and muscle strength are essential parts of hopping protocols [38]. In that regard, Ulrich et al. [39] pointed to the greater impact of short-term and explosive training compared to traditional functional training on activities such as soccer [39].

In terms of comparing the strength of the muscles of the lower extremity, positive effects were found for both groups with a higher effect for BSG than HG. For instance, the Figure 8 hop and single-leg triple hop tests, which require consecutive jumps and landing movements, were improved after eight weeks of training.

Nonetheless, HG showed higher values for vertical jump than BSG. A possible justification may be related to many hops and jumps in the hopping training, thus presenting a greater impact on this factor. The hopping training has been included as a sport-specific exercise and as a drill designed to improve speed, agility, and aerobic conditioning [40,41] and is appreciated by players and coaches increasing player compliance and participation [42,43,44]. The available evidence suggests that hopping exercises can elicit change in the stiffness of various elastic components of the muscle-tendon complex of plantar flexors [45]. The hopping training improves the stretch-shortening cycle of muscle function and induces numerous positive changes in the neural and musculoskeletal systems, and in athletic performance [46]. Additionally, the hopping exercise has been shown to enhance neuromuscular fitness and lower extremity muscle strength, which Kramer et al. [38] pointed out in their research. One of the possible reasons for advancing the functional characteristics after participating in HG is to improve neuromuscular adaptation and increase the strength of the muscles involved in lateral hop, Figure 8 hop, and single-leg triple hop protocols [39]. Another possible reason might be the strength of the joints and their construction muscles to stabilize the lower extremity joints with deep receptor activity and neuromuscular control to maintain balance when jumping in different directions. BSG training that also includes strength training with improving the ankle joint forces improves ankle strength scores by increasing plantarflexion and dorsiflexion performance as well as strengthening the ligaments around the ankle, which reduced the pressure applied to the ankle joint and improved the scores obtained in the ankle and foot ability questionnaire in individuals with CAI.

Consistent with the previous results, McKeon et al. [47] showed that performing four weeks of dynamic balance training can improve self-reported function measured by the FAAM and FAAMSPORTS of non-athletes suffering CAI [47]. Webster and Gribble also systematically reviewed functional rehabilitation interventions for CAI and stated that using closed chain rehabilitation training for four to six weeks and three to four sessions per week significantly improved the self-reported function of people with CAI [48]. One of the reasons for the improvement in the self-reported function of the participants in the neuromuscular training is due to the reduction of the limitations on the sensorimotor system as a result of these protocols. Following the results of previous studies that applied hopping protocols [10,11], the present study showed improvements on FAAM and FAAMSPORT. However, our results also revealed that participants from both groups still report symptoms of CAI [49]. Thus, more studies are needed to confirm the present findings.

It is relevant to highlight that both training protocols were applied in same days of the regular soccer training sessions. Regular soccer training included endurance sessions with light and long-term activities (such as jogging and long-distance running on the grass). Gradually, interval training was implemented during the eight-week period. Nonetheless, the present study did not address the intensity nor the exercises performed during the study period by both teams from where participants were recruited, which is a limitation that should be considered in future studies. For example, future research can employ body composition and internal and external load variables to address soccer-specific training, as well as consider other key factors, such as playing position and player’s starting status [50,51].

Furthermore, other limitations of the present research must be acknowledged. Although we present a sample power calculation, our sample is small, and it originates from two different teams. For that reason, the results must be carefully interpreted. In addition, players with bilateral ankle instability were not excluded, and the testing was simply performed on the ankle joint with the lower CAIT score, which reinforces that future studies should assess both limbs instead of only the dominant limb. The number of sets and repetitions for load progression were balanced between protocols; even so, and considering the soccer training skills, we believe a comparison of both training protocols is helpful for coaches and their staff. In addition, it was not possible to access the CG in the same week of the experimental groups to not change training routines of the players, which could affect some results, although it is possible to confirm that their physical routines were kept. Moreover, we could not confirm the reliability of the tests applied, which should be considered in future studies. Finally, the protocols only have a duration of eight weeks during the pre-season. It would be beneficial for future studies to test longitudinal approaches to access the long-term effects of the interventions and to test them during in-season periods.

Beyond the previous suggestion, future studies can use different tests and training monitoring practices to provide more knowledge on the training protocols applied. For instance, the traditional squat, counter movement, or drop jumps can be easily added. A change of direction and cardiorespiratory tests would also be beneficial to amplify the results. Lastly, the combination of body composition with internal training load and global positioning system variables may also consider other key factors, such as playing position, player’s starting status, or type of week [52].

From a practical point of view, the training protocols of this study lasted 45–60 min for BSG and 30 min for HG. Balance plus strength training seems to be very long; however, there were previous studies with such durations [21,53]. Depending on the time available for training, coaches and/or strength and conditioning professionals can choose the protocol that better suits their context.

## 5. Conclusions

This study showed that both BSG and HG improved the balance, strength, and function of soccer players with CAI. However, the BSG protocol seems to be more effective because it emphasizes each of these factors separately. It can be more effective than hopping training (which has strength and balance in nature) and has a greater impact on faster recovery for soccer players with CAI.

## Figures and Tables

**Figure 1 healthcare-10-01822-f001:**
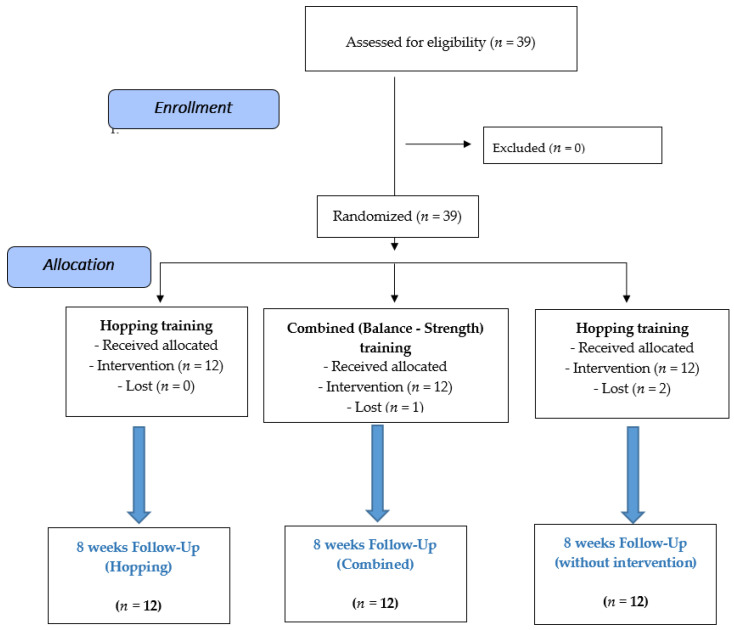
Transparent reporting of trials.

**Figure 2 healthcare-10-01822-f002:**
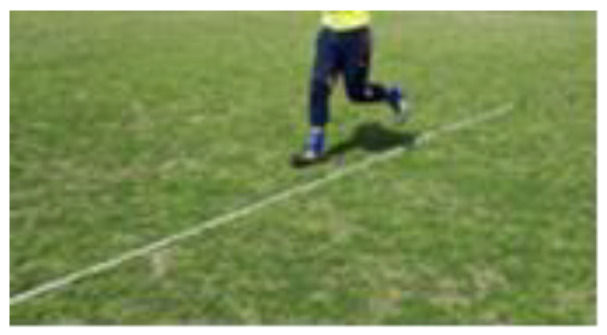
Triple hop test.

**Figure 3 healthcare-10-01822-f003:**
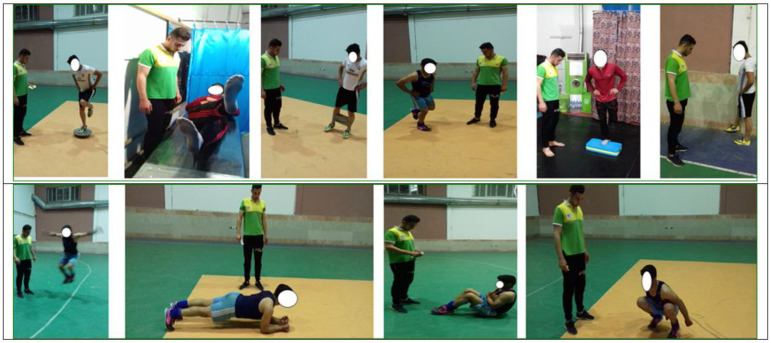
Examples of the balance plus strength exercises.

**Figure 4 healthcare-10-01822-f004:**
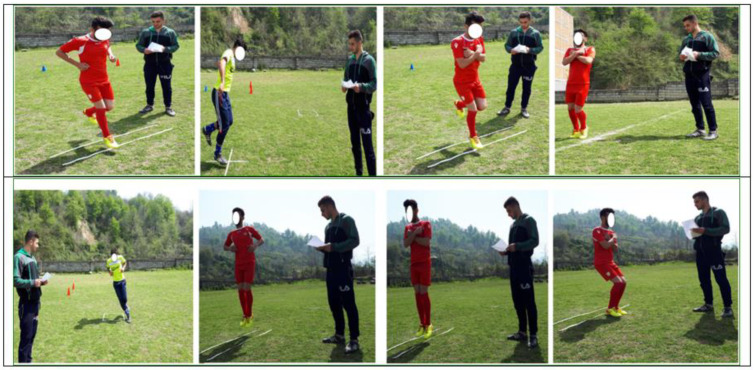
Examples of hopping exercises.

**Figure 5 healthcare-10-01822-f005:**
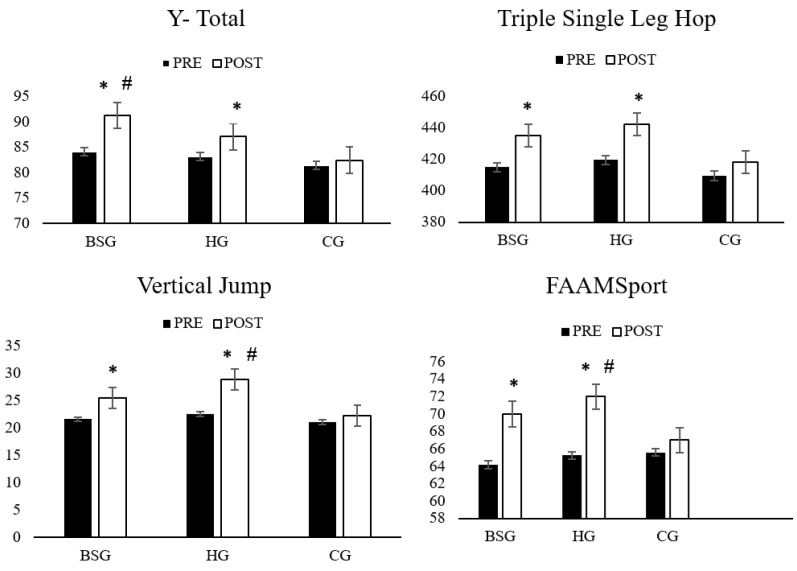
Pre to post test of Y- Total, Triple single leg hop, Vertical Jump and FAAMsport. * denotes difference between pre to post test (*p* < 0.05). # denotes difference between BSG versus HG (*p* < 0.05).

**Table 1 healthcare-10-01822-t001:** Balance plus strength protocol.

Exercise	Period of Training at the Week							
	1–2 Weeks		3–4 Weeks		5–6 Weeks		7–8 Weeks	
	Set	R/S	Set	R/S	Set	R/S	Set	R/S
Bridge to plank (s)	3	20	3	25	4	30	5	30
60-degree trunk flexion (s)	3	20	3	25	4	30	5	30
Scissor movement (R)	3	10	3	15	4	15	5	15
Working with resistance band ^a^ (R)	3	10	3	15	4	15	5	15
Movement by wall ^b^ (R)	3	10	3	15	4	15	5	15
Single leg squat	3	10	3	15	4	15	5	15
Standing on one foot (s)	3	8	3	12	4	15	5	15
Working with a wobble board ^c^ (S)	3	8	3	12	4	15	5	15
Squat Jump (R)	3	8	3	12	4	15	5	15
Tuck jump (R)	3	8	3	12	4	15	5	15
Longitudinal jump (R)	3	8	3	12	4	15	5	15
Lateral jump (R)	3	8	3	12	4	15	5	15

S = Second, R = Repetition; ^a^ flexion, extension, abduction, adduction hip; ^b^ abduction and adduction hips, inversion ankle; ^c^ balance training (standing).

**Table 2 healthcare-10-01822-t002:** Hopping protocol.

Weeks	Number of Hops	Type of Exercise	Set × Repetition
1	70	Side hop with two legs (hands-free)	3 × 10
		Forward and backward hop (hands-free)	2 × 10
		Forward hop with two legs (hands-free)	2 × 10
2	90	Side hop with two legs (hands-on chest)	2 × 15
		Forward and backward hop (hands-free)	2 × 10
		Forward hop with two legs (hands-free)	2 × 10
		Side hop with one leg (hands-free)	5 × 4
3	100	Side hop with one leg (hands-on chest)	3 × 10
		Forward and backward hop with one leg (hands-free)	2 × 10
		Forward hop with two legs (hands-on chest)	2 × 10
		Zigzag hop with two legs (hands-free)	2 × 10
4	110	Side hop with one leg (hands behind)	2 × 10
		Forward and backward hop with one leg (hands-on chest)	2 × 10
		Forward hop with on leg (hands-free)	2 × 10
		Zigzag hop with one leg (hands-free)	2 × 10
		Square hop with two legs (hands-free)	2 × 10
5–6	120	Side hop with one leg (hands behind)	2 × 10
		Forward hop with on leg (hands behind)	2 × 10
		Forward and backward hop with one leg (hands behind)	2 × 10
		Zigzag hop with one leg (hands-on chest)	2 × 10
		Square hop with one leg (hands-free)	2 × 10
		Figure 8 hop with two legs (hands-free)	2 × 10
7–8	130	Side hop with one leg (hands behind)	3 × 10
		Forward hop with on leg (hands behind)	2 × 10
		Forward and backward hop with one leg (hands behind)	2 × 10
		Zigzag hop with one leg (hands behind)	2 × 10
		Square hop with one leg (hands-on chest)	2 × 10
		Figure 8 hop with one leg (hands-free)	2 × 10

**Table 3 healthcare-10-01822-t003:** Characteristics of the study groups at baseline (mean ± SD).

Variables	BSG (*n* = 12)	HG (*n* = 12)	CG (*n* = 12)	*p*-Value
Age [years]	21.08 ± 1.78	20.83 ± 1.80	20.58 ± 1.37	0.76
Body weight [kg]	76.41 ± 9.21	75.41 ± 9.62	71.83 ± 7.60	0.42
Body height [cm]	177.58 ± 7.15	176.58 ± 5.59	178.58 ± 2.88	0.60
BMI [kg·m^−2^]	24.20 ± 0.71	24.18 ± 0.71	21.13 ± 0.13	0.15
Dominant leg length [cm]	100.41 ± 5.93	101.16 ± 3.95	98.50 ± 3.20	0.34
Activity history [years]	6.16 ± 2.62	6.08 ± 2.67	6.25 ± 2.37	0.98

BMI: body mass index; BSG: Balance-Strength training; HG: Hopping training; CG: control group.

**Table 4 healthcare-10-01822-t004:** Training effects (with CI, 95%) for the balance and function variables between groups.

Groups	Pre Test (Mean ± SD)	Post Test (Mean ± SD)	Pre vs. Post Performance Change (%)	Pre vs. Post ES (CI, 95%)
**Static balance Bass–stick (s)**				
BSG	9.33 ± 1.85	15.18 ± 2.64 ^c^	↑ 62.70	2.56 (1.50 to 3.63) ***
HG	10.07 ± 2.24	11.22 ± 2.13	↑ 11.42	0.52 (−0.24 to 1.39) *
CG	8.85 ± 1.57	8.97 ± 1.73	↑1.35	0.07 (−0.74 to 0.86)
**Y balance–Anterior (cm)**				
BSG	82.35 ± 6.18	88.56 ± 5.21	↑ 7.54	1.08 (0.70 to 2.53) ***
HG	82.41 ± 6.30	83.76 ± 5.64	↑ 1.63	0.22 (−0.57 to 1.03) *
CG	78.20 ± 6.23	78.97 ± 3.11	↑ 0.98	0.09 (−0.65 to 0.96)
**Y balance–Posteromedial (cm)**				
BSG	87.42 ± 6.63	95.71 ± 4.21 ^c^	↑ 9.48	1.49 (0.65 to 2.47) ***
HG	86.59 ± 4.37	92.27 ± 4.67 ^a^	↑ 6.55	1.25 (0.15 to 1.84) ***
CG	87.81 ± 2.78	88.63 ± 2.81	↑ 0.93	0.29 (−0.58 to 1.03) *
**Y balance–Posterolateral (cm)**				
BSG	82.54 ± 4.33	89.69 ± 4.01 ^a^	↑ 8.66	1.71 (1.09 to 3.05) ***
HG	80.69 ± 8.09	85.16 ± 6.20 ^a^	↑ 5.53	0.62 (−0.19 to 1.45) **
CG	78.19 ± 3.79	79.75 ± 5.46	↑ 1.99	0.33 (−0.49 to 1.12) *
**Y balance–total (cm)**				
BSG	84.10 ± 3.19	91.32 ± 2.33 ^c^	↑ 8.58	2.58 (1.28 to 3.31) ***
HG	83.23 ± 4.90	87.07 ± 4.23 ^a^	↑ 4.61	0.83 (0.18 to 1.87) **
CG	81.40 ± 1.33	82.45 ± 1.85	↑ 1.28	0.65 (−0.52 to 1.09) **
**Triple single leg hop (cm)**				
BSG	414.99 ± 8.40	434.98 ± 12.67 ^b^	↑ 1.28	1.85 (1.22 to 3.22) ***
HG	419.63 ± 14.49	442.22 ± 13.56 ^b,d^	↑ 5.38	1.60 (1.03 to 2.97) ***
CG	409.44 ± 15.31	418.15 ± 18.95	↑ 2.12	0.50 (−0.33 to 1.33) *
**Figure 8 hop (s)**				
BSG	9.00 ± 0.89	7.64 ± 0.34 ^b,d^	↓ 15.11	2.01 (−0.93 to 0.68) ***
HG	9.11 ± 1.11	7.03 ± 0.61 ^c,f^	↓ 22.83	2.32 (7−3.82 to–1.63) ***
CG	9.76 ± 0.70	8.88 ± 1.02	↓ 9.01	1 (−1.66 to–0.79) ***
**Vertical jump (cm)**				
BSG	21.55 ± 2.21	25.46 ± 1.93 ^b,d,h^	↑ 18.14	1.88 (1.61 to 3.78) ***
HG	22.57 ± 1.40	28.89 ± 1.71 ^c,f^	↑ 28.00	4.04 (2.01 to 4.38) ***
CG	21.04 ± 3.22	22.19 ± 3.20	↑ 5.46	0.35 (−0.34 to 1.28) *
**FAAM**				
BSG	76.16 ± 3.35	82.33 ± 3.89 ^a,d^	↑ 8.10	1.69 (0.80 to 2.66) ***
HG	76.25 ± 5.04	83.16 ± 3.88 ^b^	↑ 9.06	1.53 (0.95 to 2.86) ***
CG	75.41 ± 4.05	77.16 ± 3.48	↑ 2.32	0.46 (−0.40 to 1.21) *
**FAAMSPORT**				
BSG	64.16 ± 1.52	70.00 ± 16.51 ^c,d,i^	↑ 9.10	0.49 (−0.33 to 1.30) *
HG	65.25 ± 2.34	72.00 ± 8.59 ^c^	↑ 10.34	1.07 (0.24 to 1.95) ***
CG	65.58 ± 2.06	67.00 ± 4.60	↑ 2.16	0.39 (−0.42 to 1.20) *

SD: standard deviation; CI Confidence Interval; BSG: Balance-Strength training; HG: Hopping training; CG: control group; * Small effect size; ** Medium effect size; *** Large effect size; ↓ Decrease; ↑ Increase; ^a^ Denotes significant difference from pre to post training (*p* < 0.05). ^b^ Denotes significant difference from pre to post training (*p* < 0.01). ^c^ Denotes significant difference from pre to post training (*p* < 0.001). ^d^ Denotes significant difference with the CG post training (*p* < 0.05). ^f^ Denotes significant difference with the CG post training (*p* < 0.001). ^h^ Denotes significant difference with the HG (*p* < 0.01). ^i^ Denotes significant difference with the HG post training (*p* < 0.001).

**Table 5 healthcare-10-01822-t005:** Training effects for the strength variables between pre-post intervention and groups.

Groups	Pre Test (Mean ± SD)	Post Test (Mean ± SD)	Pre vs. Post Performance Change (%)	Pre vs. Post ES (95% CI)
**Hip abduction (Kg)**				
BSG	11.75 ± 1.74	14.79 ± 0.82 ^c,j,f^	↑ 25.87	2.23 (0.98 to 2.90) ***
HG	12.57 ± 1.87	13.58 ± 1.99	↑ 8.03	0.52 (−0.27 to 1.35) *
CG	11.73 ± 1.03	11.91 ± 0.97	↑ 1.53	0.17 (−0.68 to 0.92)
**Hip adduction (Kg)**				
BSG	10.51 ± 1.25	13.19 ± 1.36 ^c,j,f^	↑ 25.49	2.05 (1.22 to 3.23) ***
HG	10.46 ± 1.66	12.14 ± 1.02 ^b^	↑ 16.06	1.21 (0.58 to 2.37) ***
CG	10.67 ± 0.93	10.83 ± 0.88	↑ 1.49	0.17 (−0.68 to 0.92)
**Knee flexion (Kg)**				
BSG	11.49 ± 0.72	13.56 ± 0.82 ^c,j,f^	↑ 18.01	2.68 (1.32 to 3.37) ***
HG	11.79 ± 1.27	13.61 ± 1.43 ^c,f^	↑ 15.43	1.34 (0.69 to 2.52) ***
CG	11.36 ± 0.77	11.58 ± 0.76	↑ 1.93	0.28 (−0.59 to 1.01) *
**Knee extension (Kg)**				
BSG	11.82 ± 1.56	12.56 ± 1.42 ^d^	↑ 6.26	2.50 (−0.17 to 1.47) ***
HG	11.45 ± 1.55	13.58 ± 0.91 ^c,f^	↑ 21.57	1.67 (0.75 to 2.59) ***
CG	11.17 ± 0.79	11.38 ± 0.38	↑ 1.88	0.33 (−0.62 to 0.99) *
**Ankle plantarflexion (Kg)**				
BSG	8.93 ± 1.07	11.30 ± 1.07 ^c,d,j^	↑ 26.53	2.21 (1.46 to 3.58) ***
HG	9.24 ± 1.23	11.90 ± 1.23 ^c,f^	↑ 27.13	2.16 (1.29 to 3.33) ***
CG	9.36 ± 0.79	9.66 ± 0.71	↑ 3.20	0.33 (−0.51 to 1.10) *
**Ankle dorsiflexion (Kg)**				
BSG	8.20 ± 0.73	10.91 ± 1.18 ^c,d^	↑ 33.04	2.76 (10.91 to 8.2) ***
HG	8.71 ± 0.73	11.41 ± 1.28 ^c,f^	↑ 24.69	2.59 (11.41 to 8.71) ***
CG	9.15 ± 0.68	9.42 ± 0.66	↑ 2.95	0.40 (9.42 to 9.15) *
**Ankle eversion (Kg)**				
BSG	4.69 ± 0.47	6.09 ± 0.72 ^c,f^	↑ 29.85	2.30 (0.91 to 2.80) ***
HG	4.95 ± 0.52	5.48 ± 0.71	↑ 10.70	0.85 (0.04 to 1.72) **
CG	4.70 ± 0.48	4.87 ± 0.51	↑ 3.61	0.34 (−0.53 to 1.08) *
**Ankle inversion (Kg)**				
BSG	5.53 ± 0.54	7.45 ± 0.51 ^c,f,h^	↑ 34.71	3.65 (2.55 to 5.20) ***
HG	5.32 ± 0.74	6.49 ± 0.75 ^c,d^	↑ 21.99	1.57 (0.85 to 2.73) ***
CG	5.52 ± 0.63	5.69 ± 0.72	↑ 3.07	0.25 (−0.57 to 1.04) *

SD: standard deviation; CI Confidence Interval; BSG: Balance-Strength training; HG: Hopping training; CG: control group; ↓ Decrease; ↑ Increase; ***** Small. ****** Medium. ******* Large. ^b^ Denotes significant difference from pre to post training (*p* < 0.01). ^c^ Denotes significant difference from pre to post training (*p* < 0.001). ^d^ Denotes significant difference with the CG post training (*p* < 0.05). ^f^ Denotes significant difference with the CG post training (*p* < 0.001). ^j^ Denotes significant difference with the HG (*p* < 0.05). ^h^ Denotes significant difference with the HG (*p* < 0.01).

## Data Availability

The data presented in this study are available on request from the corresponding author.

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
