# Peer review of "What Does Provide Better Effects on Balance, Strength, and Lower Extremity Muscle Function in Professional Male Soccer Players with Chronic Ankle Instability? Hopping or a Balance Plus Strength Intervention? A Randomized Control Study"

_healthcare, 2022, doi:10.3390/healthcare10101822_

Round 1

Reviewer 1 Report

Dear Authors,

Substantial revisions need to be made to ensure this is publication standard.

See specific recommendations as follows:

Abstract:

Lines 26-7: Please improve the clarity of this sentence. Dynamometer used to measure strength of the muscles around the hip, knee and ankle joints? “The Bass stick measure static balance and the Y balance test dynamic balance”?

Lines 27 -28: ‘functional tests were carried out’?

Line 28: A repeated measures ANOVA…

Line 30: “The BSG…”

Line 31: “…while the HG…” “… with small to large effect sizes (ES)”

Line 36: “the hopping protocol” not ‘hooping’.

Introduction:

Line 46: Please specify the ‘return rate’. Is this return to training or full competition?

Line 47: “require” works better here than ‘request’

Line 47: Is this best clarified as ‘change in direction’?

Line 53: The first article here calls it ‘postural balance’ whereas the second ‘dynamic posture-control’ but you introduce it as ‘dynamic dimension’. Can you clarify this further?

Line 56: Please refine this sentence. An injury occurring in the structural ligaments’ strength does not make sense.

Line 59: “…mechanical receptor injuries associated with CAI can lead to functional and balance issues…”

Line 64: Can you be more direct with these statements? I suggest remove ‘in their results regarding the investigation’ and change to “Shiftan et al observed improvement from a proprioception training program following CAI injury”

Lines 67-70: Suggest separating this long sentence into two for clarity.

Line 71: ‘Another training approach’ ‘is in the use of hopping protocols’

Line 80: ‘Literature is scarce..’

Lines 82-84: Long sentence. Consider breaking into two more direct sentences.

Line 84: “The aim was..”

Methods

Line 101-112: This is a very long sentence. Consider making this more readable by shorter sentences.

Line 103: Consider just referring to the IAC criteria rather than listing each point in turn.

Line 126: “..while a score between 0 and 27 indicates more severe ankle instability”

Line 126: This is quite a wide range of ‘more severe’!

Line 155: Remove this sentence about baseline differences as it is in your results.

Lines 157-160: Is this a repeat of the information above?

Line 161: “exercises” not exercise

Line 165: Was this static or dynamic calf stretching?

Line 168: “the CG”

Line 174: Please define further what is meant by ‘within-team match’

Line 176: “lasted eight weeks”

Line 177: “the CG”

Line 185: ‘… by the dominant leg, which was specified using the ball kicking test..”

Line 200: Remove the “:”

Lines 199-214: Where are the references outlining the procedure for the Y-balance test?

2.6.1 and 2.6.2: Please outline what instructions were given to ensure appropriate technique (e.g. do not bend forward from the waist whilst undertaking a single leg squat etc).

2.6.2 Statistical analysis should be ‘2.6.3 Statistical Analysis”

Discussion

Line 372: Should be singular “balance” rather than ‘balances’

Lines 373-4: Avoid using terms such as ‘probably better results’ as this is conjecture. Your statements should be using appropriately scientific terms.

Line 375: Performing exercises that address ‘all the neuromuscular variables’ could be timely and have issues in terms of adherence/maintenance. How confident are you that each of the prescribed exercises in the BSG group are warranted?

Line 387-388: Avoid general terminology such as ‘in the same line’. You need to be specific here.

Line 398: As above comment on general linkages to previous studies. Be specific.

Line 399: Change from ‘Other relevant variable’ as this needs to be clearer.

Lines400-403: Please revise this sentence as it does not read well.

Lines 410-412: Please do not speculate. Stick to your findings only.

Line 417: “…the body triggering the core muscles of the body…”

Line 428: What do you mean by ‘combination bridge’?

Line 446: How does hopping training enhance NM fitness and lower extremity muscle strength?

Line 464: Again do not speculate. Draw from your results and link these with previous findings.

Line 489: Why could reliability not be confirmed?

Lines 505-506: Whilst the BSG Protocol appears to be more effective, I am still not convinced that each of the exercises are necessary, essentially leading to extra training time that could be focused elsewhere. Until separate analyses of each individual exercise’s effect and then in combination is carried out I would be careful with the language here around effectiveness.

Line 509-510: Why would a coach choose a potentially inferior protocol when the current protocol still needs considerable validation?

Author Response

Thank you for good comments, I attached file.

Reviewer 2 Report

Dear authors,

The aim of this study was to the effects of eight weeks of hopping protocol and a combined protocol of balance plus strength in a within-between group analysis. It is an interesting manuscript with a robust research design which can be accepted after minor revisions.

Please, consider the attached revisions in order to further improve the quality of the manuscript.

We look forward to hearing from you.

Good work and kind regards.

Author Response

Thank you for good comments. I attached file.

Round 2

Reviewer 1 Report

Dear Authors,

Thank you for addressing my concerns. The manuscript is much improved.